# An Interactive Feeder to Induce and Assess Emotions from Vocalisations of Chickens

**DOI:** 10.3390/ani14091386

**Published:** 2024-05-06

**Authors:** Antonis Golfidis, Buddhamas Pralle Kriengwatana, Mina Mounir, Tomas Norton

**Affiliations:** Faculty of Bioscience Engineering, Katholieke Universiteit Leuven (KU LEUVEN), Kasteelpark Arenberg 30, 3001 Leuven, Belgium; antonis.gkolfidis@kuleuven.be (A.G.); mina.mounir@kuleuven.be (M.M.)

**Keywords:** vocal emotions, animal–computer interaction, laying hens, chickens, vocalisations, birds

## Abstract

**Simple Summary:**

The vocalisations that an animal produces could provide a window into its emotional state. More knowledge on how emotional states are expressed in the vocalisations of birds could even be used to improve the welfare of farmed poultry. The present study introduced a novel device designed to trigger different emotional states in hens using different physical and chemical stimuli. The final device was able to elicit a broad range of vocalisations without any human interference. It was found that the hens not only actively used the device as a feeder but also responded vocally to both positive and negative stimuli. Preliminary findings indicate that the vocal responses of the hens vary as a function of the intensity of the emotional state they experience. This research contributes to understanding poultry emotions and has the potential to open new opportunities in understanding and improving their welfare in farming environments.

**Abstract:**

Understanding the emotional states of animals is a long-standing research endeavour that has clear applications in animal welfare. Vocalisations are emerging as a promising way to assess both positive and negative emotional states. However, the vocal expression of emotions in birds is a relatively unexplored research area. The goal of this study was to develop an interactive feeding system that would elicit positive and negative emotional states, and collect recordings of the vocal expression of these emotions without human interference. In this paper, the mechatronic design and development of the feeder is described. Design choices were motivated by the desire for the hens to voluntarily interact with the feeder and experience the different stimuli that were designed to induce (1) positive low-arousal, (2) positive high-arousal, (3) negative low-arousal, and (4) negative high-arousal states. The results showed that hens were motivated to engage with the feeder despite the risk of receiving negative stimuli and that this motivation was sustained for at least 1 week. The potential of using the interactive feeder to analyse chicken vocalisations related to emotional valence and arousal is being explored, offering a novel proof of concept in animal welfare research. Preliminary findings suggest that hens vocalised in response to all four stimulus types, with the number of vocalisations, but not the probability of vocalising, distinguishing between low- and high-arousal states. Thus, the proposed animal–computer interaction design has potential to be used as an enrichment device and for future experiments on vocal emotions in birds.

## 1. Introduction

The welfare of captive non-human animals (hereafter, animals) under human care is a growing public concern, with both direct and indirect impacts on human health and well-being and environmental sustainability. Recent scientific interest in the role of emotional states in animal welfare is motivated by the idea that good welfare should not only be characterised as a lack of negative states but also include the opportunity to experience positive emotional states [1,2,3]. In short, animals should not only survive, but thrive in environments that provide a balance between what an animal likes to do and what is healthy for it to do. Thus, knowing when an animal is in a positive emotional state remains a critical and ongoing research endeavour in animal welfare sciences.

The scientific study of animal emotions is flourishing, being supported by different conceptual frameworks. These frameworks can enable cross-species comparisons of emotional states and are complemented by technological and methodological advancements that facilitate the observation and quantification of these states [4,5]. Emotions can be defined as internal, short-lived psychobiological states that are reactions to specific internal or external objects or events that are biologically or ecologically relevant to an individual. Emotions are conceptually different from moods, which are more persistent internal states that are not necessarily triggered by a specific object or event [5]. Animal emotions can be viewed as consisting of two dimensions: arousal and valence [4]. Arousal refers to the intensity of the emotion while valence refers to the hedonic value of the emotion (positive or negative). Although there is no access to animals’ subjective emotional experiences, they are likely accompanied by changes in arousal and valence that can be measured as changes in behaviour, cognitive processing, physiology, and neural activity [4,5].

Vocalisations are emerging as promising non-invasive measures of animal emotional states [6]. Vocalisations are produced by almost all vertebrates and may be reliable markers of emotions because the midbrain and limbic systems that process emotional stimuli also play a role in vocal production [7,8,9]. In vertebrates, the arousal response includes elevation of sympathetic nervous system activity, which leads to a higher respiratory rate and muscle tension, both of which modulate acoustic features of vocalisations [10].

Research into acoustic features that distinguish vocalisations produced in positive and negative emotional states shows that across several mammalian species, vocalisations produced in positive emotional states tend to be shorter and have a lower fundamental frequency (although species differences exist [6,7,11,12]). For instance, horse whinnies produced during positive situations (social reunion) were lower pitched and shorter in duration than whinnies produced during negative situations (social isolation; [13]). Similarly, grunts, screams, and squeals of wild boars during positive situations (food anticipation and affiliative interactions) were shorter in duration with less amplitude modulation and lower frequency than in negative situations (agonistic interactions) [14]. Compared to mammals, little progress has been made on the identification of vocal indicators of positive emotions in birds. Research on vocal emotions in birds is highly concentrated on negative emotions, with a large bias towards high-arousal calls that signal current levels of pain, fear, or distress [6,7,15].

The aim of this study was to design a device capable of stimulating domestic layer hens to vocalise across different levels of emotional arousal and valence. To the best of the authors’ knowledge, no existing device has been able to induce vocalisations across all four quadrants of the dimensional model of emotions, i.e., positive low-arousal, positive high-arousal, negative low-arousal, and negative high-arousal emotional states. The design was inspired by findings of two studies on vocalisations during positive emotional states in domestic chickens, which observed variations in call types—namely food calls, fast clucks, and gakels—based on the anticipation of rewards [16,17]. While [17] found that the peak frequency increased with the arousal level in positively valenced situations, neither study analysed whether the acoustic features of these calls varied between the reward and no-reward conditions. Addressing this gap, the device is designed to explore how emotional valence and arousal are encoded in vocalisations, offering new insights for poultry welfare monitoring by assessing the birds’ emotional states through their calls.

The ability to measure valence and arousal is essential for linking vocalisations to specific emotional states. Given the challenge of assessing an animal’s emotional valence due to the absence of direct non-invasive methods, lack of verbal communication, and the inherent subjectivity in labelling emotional states, a feeder was developed to provide a range of stimuli known to impact approach or avoid behaviours. To quantify the intensity of an emotion, behavioural and physiological indicators that correlate with levels of arousal can be measured by devices such as heart rate sensors, thermal cameras, and RGB cameras. Therefore, the setup should accommodate the requirements these devices need to capture data.

Positioned within a broader research framework, the objective is to distinguish between vocalisations based on emotional valence and arousal, necessitating a system that prompts vocal responses to both positive and negative stimuli. This paper is structured around two main objectives: the development and testing of an interactive feeder as a novel method for eliciting and analysing chicken vocalisations, serving as a proof of concept in the intersection of animal welfare and technology. To this end, various automated feeder designs are discussed while focusing on refining the optimal design to facilitate this study’s goals. Key aspects of the feeder’s usability are investigated, such as the birds’ ability to learn how to activate the feeder, their engagement frequency, and the consistency of daily usage patterns. Furthermore, audio data from two trials are analysed to explore the likelihood and frequency of vocalisations produced by hens while interacting with the feeder. The findings presented offer preliminary insights into the potential of such interactive devices in enhancing our understanding of animal emotional states, setting a foundation for further exploration in this novel intersection of fields.

## 2. Materials and Methods

### 2.1. Feeder

#### 2.1.1. Stimuli

The objective was to create a device that hens could voluntarily activate to deliver stimuli inducing positive and negative emotional states of different arousal intensities (low and high). Therefore, the design needed to deliver the selected stimuli in a suitable manner. Stimuli were chosen to induce low- and high-arousal states, with each assigned as positive or negative based on the previous literature linking these stimuli with approach or avoid behaviours [18,19,20,21,22,23]. The rationale for using approach and avoid behaviours as proxies of emotional valence was based on the assumption that, in general, animals will move towards stimuli that are positive and away from stimuli that are negative. Hens are required to initially approach the device to activate it. Upon activation, stimuli—positive or negative—are randomly dispensed. Thus, the birds do not know whether they will receive a positive or negative stimulus until they approach and activate the feeder. Positive stimuli such as rice and mealworms generally encourage the birds to continue approaching the corresponding container, whereas negative stimuli lead to avoidance behaviour.

The selected positive stimuli were food items, including mealworms and rice [18]. These choices were based on preliminary observations indicating that these foods elicited stronger engagement compared to other options like corn and peanuts. Rice was chosen instead of corn (which was used in [18]) because pilot data showed that when given a choice between rice, corn, mealworms, and peanuts simultaneously, hens chose to consume mealworms and rice first and in greater quantities when compared to corn or peanuts. Negative stimuli included two puffs of air to the face (one every 30 s for 1 min), four puffs of air to the face (one every 15 s for 1 min) [20,23], and quinine-coated rice derived from 1% and 4% suspensions of quinine (a bitter tasting substance) dissolved in water that was mixed with rice [21,22]. Thus, it was assumed that mealworms and rice would trigger a positive valence of varying arousal levels, and rice + 1% quinine and rice + 4% quinine would trigger negative valence of varying arousal levels. Food colouring was used to make the rice with 1% quinine green and the rice with 4% quinine blue. This was based on the hypothesis that hens could differentiate between these conditions and learn to associate the colours with the different levels of bitterness. To motivate consistent engagement, the device was first intended to act as the hens’ primary source of food. The device was designed as a feeder with the capacity to dispense 7 different types of stimuli to the hens (Table 1).

In trial 1, regular feed was given 80% of the time the device was activated, and positive and negative stimuli could each be given 10% of the time (5% for mealworms, 5% for rice, 2.5% for rice + 1% quinine, 2.5% for rice + 4% quinine, 2.5% for two air puffs, and 2.5% for four air puffs). Hens were provided with ad libitum access to regular feed in a separate container to ensure that they received enough food during the first few days before they learned how to use the feeder. However, visual observations during the first few days of trial 1 indicated that hens frequently interacted with the device even though regular feed was available. Thus, the regular feed was left in a separate container throughout the trial. In trial 2, the separate container with regular feed was maintained for the duration of the trial, but the probability of positive stimuli was increased. Positive stimuli were provided 95% of the time (85% for mealworms, 10% for rice), and negative stimuli 5% of the time (only air puffs, with 2.5% for two air puffs and 2.5% for four air puffs), to enhance the sample size for vocal responses to positive stimuli.

#### 2.1.2. Design Ideas

In addition to dispensing food and air puffs, the design of the device should allow for the measurement of arousal levels to validate the arousal level of animals at the time of vocalisation. To non-invasively measure arousal, a thermal camera was proposed to record the surface temperature of the head (face, eye, comb, and wattle) from a profile view, as this has been shown to change in response to both positive and negative events and different magnitudes of stress [23,24,25,26]. Considering all the specifications, three potential feeder designs were conceptualised. Each design has advantages and disadvantages that are summarised in Table 2.

Design 1: The tower design (Figure 1) features a central container with compartments for different food stimuli, equipped with a sensor and air puff outlet at hen height. Upon activation, food is delivered from individual compartments into a central feeding area controlled by motorised panels. After the programmed amount of food is dispensed, the birds can eat in the feeding area for a set amount of time. Once this time has expired, a motorised trap door on the floor of the feeding area opens, driving uneaten food in the waste disposal container.

The tower design has an advantage that hens remain within a limited area, which would increase the probability of capturing usable images with a thermal camera, while minimising the risk of spilling food. However, some practical concerns arise with this design as it necessitates a complex mechanism to support the stored food’s weight, only dispensing a controlled portion at a time. Additionally, it requires six motors in total: five for various food options and one dedicated to food disposal. Lastly, the potential mixing of uneaten food could lead to waste, and the placement of the sensor might require hens to stretch their necks.

Design 2: The tube design (Figure 2) consists of five independent 4” PVC tubes, each containing a specific type of food, covered by a lid. On the lid of the central tube, there is the proximity sensor and the air puff outlet, with the speaker mounted in the central tube itself. Each tube has its own motor—connected to the central electronics box—which lifts the lid for the specified activation period, providing access to a distinct food stimulus.

The tube design has been partially used by farmers to feed chickens while addressing the challenge of removing uneaten food and has been partially used by farmers to feed chickens. It also facilitates the use of a thermal camera. The construction of the motorised lids presents its own set of challenges, both in the intricacy of the mechanism and the need for five distinct motors. A potential concern is the hens disrupting the stimulus delivery process, either by intentionally spilling food or interfering with the lids’ operations. Lastly, the air puff outlet could be blocked by food.

Design 3: The floor rotator design (Figure 3) emerged from discussions with peers at the London Metropolitan University who explore ACI concepts [27]. The proposed system—ideally positioned on a raised platform—is designed with a multi-component structure. The upper section features a plastic surface with five holes along with a designated area for the interaction components, including the sensor, air puff outlet, and speaker. The lower section is intended to be at ground level to replicate hen foraging behaviour. This part houses five food containers for storing various food types and includes openings for wiring that connects the top section’s components with the rest of the electronics. Between these two sections is a rotatable disc, controlled by a stepper motor, with a single hole that aligns with both the upper and lower sections’ openings. To access food, hens trigger the sensor in the upper section, causing the disc to rotate and align its opening with one of the food containers, allowing the birds to feed.

The floor rotator design also bypasses the issue of uneaten food removal, while ensuring that there will be a low risk of spilling food. It also resembles the natural ground foraging behaviour, making it appealing to hens. A significant benefit lies in its electronic simplicity, anchored by a single stepper motor with some additional calibration required to fine-tune the rotation angles. Importantly, this option guarantees that the food delivery process would be hard to disrupt. Nevertheless, a couple of disadvantages remain, such as potential inconsistencies in thermal recording due to varying head angles of the hens in relation to the camera’s position, and the complexity of the elevated platform construction, especially when considering the pen’s dimensions and flooring.

#### 2.1.3. Final Design

A modified version of the floor rotator design was implemented, believed to offer compelling advantages, while also addressing some of its challenges. The device is composed of two primary components: the rotatable disc and the containers. Figure 4 shows the feeder design.

The feeder incorporates six distinct plastic containers (Figure 4A). The baseline container, which is empty of food, is the place where the hens have access while no stimulus is provided. As hens are anticipated to position their heads over this container when initiating the system, the air puff outlet is located in the baseline container. The remaining containers house the various food types. Besides these containers, the inner section of the feeder houses the stepper motor, the speaker, and the ultrasonic sensor. The sensor was fixed on the wall of the feeder (Figure 4B) so that system activation was triggered when a hen approached this particular side of the feeder.

The top layer of the device is a rotatable plastic disc, with an opening that matches the containers’ dimensions. This disc is secured to the stepper motor via a screw, ensuring synchronised rotation between the motor and the disc. Metal wheels were added to reduce friction (Figure 4D). The entire apparatus stands elevated on plastic legs, ensuring hens can access the container contents easily, instead of an elevated platform (due to pen size constraints).

The core electronic components are contained in a plastic compartment under the feeder (Figure 4E). This compartment connects the USB, power cables, wires, and tubes that are necessary for the air puff mechanism to the feeder. The air puff system comprises an air compressor, a solenoid valve, and the requisite plastic tubing connecting each component. Pigeon sticks were added to the rotating disc to prevent hens from stepping on it. (refer to Discussion for more details).

#### 2.1.4. Hardware Description

The embedded system comprises several essential components, including an Arduino Mega 2560 microcontroller, a Real-Time Clock (RTC) module, an SD card reader, an ultrasonic sensor, a miniature speaker, a pneumatic solenoid valve, an air compressor, a uStepper S microcontroller, and a Nema Stepper Motor. Figure 5 provides a visual representation of the system’s components and their interconnections. Below is a brief description of each component:(1)Arduino Mega 2560: An open-source microcontroller board serving as the central control unit responsible for coordinating the operation of the entire system.(2)HC-SR04 Ultrasonic sensor: A proximity sensor that measures distance using sound waves. It is set to detect when a hen is within 10 cm.(3)DS3231 Real-Time Clock module: The Real-Time Clock (RTC) module reports feeder activation times, measures the duration of stimuli delivery, and determines whether the feeder should be active during the daytime or inactive during the night.(4)Miniature speaker: A 0.5 W miniature speaker was used to emit a pure tone sinewave right before the offset of the stimulus presentation. It also serves to attract the attention of the hens, encouraging them to approach the feeder, and startles their reflexes, prompting them to withdraw their heads from the container as the disc returns to its baseline position.(5)Micro-SD card reader SPI interface + Micro-SD card: Essential for data storage, it logs the activation times of the feeder, including details like feeder ID, stimulus ID, date, and time. Data retrieval can be achieved in various ways, such as removing the SD card after the experiment, connecting a laptop to the Arduino via USB to view activation logs in the Serial Monitor, or using a Bluetooth module for remote data acquisition to minimise human presence.(6)Pneumatic solenoid valve: This component was integrated into the experimental apparatus to deliver controlled air puffs. Operational control of this valve is managed by the Arduino microcontroller, enabling the release of air stored in an air compressor.(7)uStepper S stepper driver: This is a microcontroller stepper driver which is compatible with Arduino boards. The two controllers, namely the uStepper S and the Arduino Mega, communicate using the I2C protocol, with the Arduino acting as the leader. This bidirectional interaction allows for the exchange of messages between the controllers. Specifically, the Arduino Mega sends instructions specifying the desired angle for the stepper motor, while the uStepper S responds by reporting the current angle at which the stepper motor is positioned. This dynamic information exchange achieves accurate and real-time control over the rotatable disc’s movements.(8)Nema 23 stepper motor: A Nema stepper motor is an electric motor which converts digital input pulses from the uStepper S into precise mechanical shaft rotation in a series of equally spaced steps. The rotation parameters such as speed, acceleration, deceleration, and the exact angles were fine-tuned to ensure the smooth operation of the feeder.

#### 2.1.5. Description of Software

The software, developed in the Arduino environment, consists of two distinct files: one tailored for the Arduino Mega board and the other for the uStepper S board. The programme’s flowchart is illustrated in Figure 6. The basic steps of the algorithm are described below:Time check: The current time is verified to ensure that feeder operations occur only during the daytime (8:00 AM to 10:00 PM).Object detection: The system continuously assesses the distance measured by the ultrasonic sensor. If an object is detected within a 10 cm range, it is interpreted as an intention to activate the feeder.Stimulus estimation: Each stimulus type has an assigned probability. The system generates a random number (ranging from 0.0 to 100.0) to determine the exact stimulus to be delivered.Data storage: Details about the chosen stimulus, along with the timestamp of its presentation, are recorded on the SD card.

Stimulus delivery:Air puff: For low- or high-intensity air puffs, the Arduino activates the solenoid valve.Food stimulus: The Arduino sends an encoded message to the uStepper S board indicating the desired angle of rotation. The stepper motor moves the rotatable disc to the defined angle, providing access to the specified food type.Sound display: After the expiration of the opening period, a simple tone is produced via the miniature speaker.System reset: Once the stimulus delivery concludes, the stepper motor returns to its initial position. Concurrently, the ultrasonic sensor initiates a new scanning cycle to detect objects.

### 2.2. Animals and Setup

Ethical approval was obtained prior to experiments from the Ethical Committee for Animal Experimentation at KU Leuven (project number 082/2023).

#### 2.2.1. Animals and Housing

Four ISA brown laying hens were obtained from TRANSfarm, KU Leuven. The hens were tested in pairs, in two separate trials. The hens were 30 weeks old at the start of trial 1 (from 7 July 2023 to 31 July 2023) and 39 weeks old at the start of trial 2 (from 12 September 2023 to 22 September 2023). For both trial 1 and trial 2, the experiments were concluded at midday on their respective final days. Due to unforeseen technical difficulties, the device did not deliver stimuli reliably until 24/07; thus, for trial 1, the hens’ data were analysed only between 24th and 31st July, which was 15 days after the feeder was first introduced to the hens. The hens were housed in pens 2.3 × 2.3 × 0.8 m (L × W × H), with a net covering the top of the pen, in a climate-controlled room at the Department of Biosystems, KU Leuven animal facility. The ambient temperature was kept at 21 °C and humidity between 60 and 70%. The daily light/dark cycle was 14 h/10 h light/dark, with lights on between 08:00 and 22:00. The hens received ad libitum food and water in conventional poultry feeders, access to a dust-bathing substrate, a pecking stone, an elevated perching area for roosting, and nest boxes (2 per pair). Daily checks were conducted to ensure that the birds had sufficient food and water and were in good health.

#### 2.2.2. Recording Setup

In each trial, two devices were placed in the pen (Figure 7). A directional microphone (AKG C 391 B) was set up above each device, and another omnidirectional one (AKG SE300 B) was used to record ambient sounds within the enclosure. All microphones were connected to a soundcard (Focusrite Clarett + 4Pre) which was then connected to a laptop that stored audio recordings. Recordings were made continuously (24 h/day). To capture video data, a camera was installed above the pen (Dahua DH-SD1A203T-GN). This camera was connected to a Network Video Recorder (Dahua DHI-NVR4208-8P-4KS2) that stored the video recordings. Recordings were made only when the lights were on (from 08:00 to 22:00).

### 2.3. Data Analysis

#### 2.3.1. Feeder Usage Statistics

To investigate the number of feeder activations across experimental days and to identify potential hourly patterns, the activation logs retrieved from the SD card of each feeder were utilised. Following each experimental trial, feeder activation data were compiled in the form of text files. These logs contained details regarding which feeder was activated, the stimulus delivered, and the exact date and time of the activation. The text files were transformed into CSV format to enable easier further analysis steps. Subsequently, activation logs were processed using R software, version 4.3.2.

#### 2.3.2. Vocalisations

One of the primary objectives of this study was to provide a proof of concept by analysing vocalisations emitted across different experimental stimuli. While audio data were continuously recorded, feeder activations were irregular. Thus, the initial step of the analysis involved extracting audio that corresponded to the feeder activation logs. A Python script facilitated the extraction of 1 min segments that include feeder activations—spanning 5 s before to 55 s after the activation (Figure 8). For each feeder, the audio channel from the directional microphone aimed at that feeder was utilised.

Subsequently, each segment was processed using Audacity 3.3.3 to generate its spectrogram. These extracted segments were then annotated to determine the onset and offset of hen vocalisations. The times when the feeder opened and closed were also marked. This allowed any variances in the timing of vocalisations to be explored across different stimuli. Every segment was saved along with its respective Audacity project file and text annotation file. A subsequent Python script was employed to generate a database, registering the beginning and end of each vocalisation, its clarity grade, and the file in which it was located, setting the stage for in-depth statistical analysis.

Statistical analyses on feeder usage and vocalisations were conducted using the R Statistical language (version 4.3.0; R Core Team, 2023) on macOS 14.1, using the packages MuMIn [28], glmmTMB [29], lubridate [30], DHARMa [31], chron [32], report [33], patchwork [34], ggplot2 [35], dplyr [36], and tidyr [37].

## 3. Results

### 3.1. Feeder Engagement

#### 3.1.1. Engagement across Days

Hens activated the feeder a total of 1958 times in trial 1 (mean ± SD = 244.75 ± 93.94 times per day) and 1311 times in trial 2 (mean ± SD = 119.18 ± 72.11 times per day). Across both trials, the total number of times the birds were exposed to normal feed (Neutral) = 1571, rice (Positive + Low Arousal) = 181, mealworms (Positive + High Arousal) = 1127, rice + 1% quinine (Negative + Low Arousal) = 86, rice + 4% quinine (Negative + High Arousal) = 105, two air puffs (Negative + Low Arousal) = 117, and four air puffs (Negative + High Arousal) = 63. Feeder engagement varied between the two trials and across days (Figure 8). A generalised linear model with a negative binomial distribution was used to ask whether there was a trend for feeder engagement to increase or decrease over time. As feeder engagement and experimental day appeared to have a nonlinear relationship in trial 2, trial 1 and trial 2 were modelled separately. For both trials, a linear regression model was compared with a polynomial regression model, selecting the most parsimonious model as the model with the lowest Akaike information criterion (AIC). For trial 1, it was observed that feeder engagement did not show significant variation across days. The model’s explanatory power was very weak (Nagelkerke’s R^2^ = 3.15 × 10^−3^). The model’s intercept, corresponding to day = 0, was at 2.90 (95% CI [2.48, 3.35], *p* < 0.001). The effect of day was statistically non-significant and positive (beta = 0.02, 95% CI [−0.07, 0.12], *p* = 0.640). On the other hand, feeder engagement varied by experimental day in trial 2. The model’s explanatory power was moderate (Nagelkerke’s R^2^ = 0.19). The model’s intercept, corresponding to day = 0, was at 1.46 (95% CI [0.61, 2.38], *p* < 0.001). The effect of day [first degree] was statistically significant and positive (beta = 0.53, 95% CI [0.20, 0.84], *p* < 0.001) while the effect of day [second degree] was also statistically significant and negative (beta = −0.05, 95% CI [−0.07, −0.02], *p* < 0.001).

#### 3.1.2. Engagement within a Day

Hens interacted with the feeder throughout the day, possibly engaging more consistently in the evening (6 PM–10 PM) than in the morning (8 AM–10 AM, Figure 9). To check whether hens showed a preference for engaging with the feeder at specific times of the day, separate negative binomial models were fitted for trial 1 and trial 2 to predict the number of feeder activations by time (divided into 2 h bins). Model fit was evaluated using linear or polynomial regressions with AIC. For trial 1, the model’s explanatory power was weak (Nagelkerke’s R^2^ = 0.04). The model’s intercept, corresponding to timebin = 0, was at 2.70 (95% CI [2.31, 3.11], *p* < 0.001). The effect of the timebin was statistically non-significant and positive (beta = 0.07, 95% CI [−0.02, 0.16], *p* = 0.135). For trial 2, the model’s explanatory power was substantial (Nagelkerke’s R^2^ = 0.40). The model’s intercept, corresponding to timebin = 8 AM, was at 2.45 (95% CI [2.28, 2.63], *p* < 0.001). Within this model the effect of timebin [first degree] is statistically significant and positive (beta = 4.63, 95% CI [2.90, 6.37], *p* < 0.001). The effect of timebin [second degree] is statistically significant and positive (beta = 2.31, 95% CI [0.64, 4.01], *p* = 0.009). Thus, the hens in trial 1 did not have a preferred time to interact with the feeder, while the hens in trial 2 preferred to interact with it more in the evening.

### 3.2. Vocal Activity

The results presented herein should be considered preliminary, highlighting the interactive feeder’s potential rather than providing definitive conclusions on emotional valence and arousal in chicken vocalisations.

#### 3.2.1. Probability of Vocalising

Manual labelling of vocalisations was conducted on 930 sound files that recorded 1 min of audio after the feeder was activated. The number of files analysed for each stimulus type were normal feed (Neutral) = 200, rice (Positive + Low Arousal) = 161, mealworms (Positive + High Arousal) = 277, rice + 1% quinine (Negative + Low Arousal) = 76, rice + 4% quinine (Negative + High Arousal) = 97, two air puffs (Negative + Low Arousal) = 77, and four air puffs (Negative + High Arousal) = 42.

Overall, it was found that the probability that hens would vocalise after activating the feeder was 51%. This value varied depending on the type of stimulus they experienced when activating the feeder (Figure 10). Hens were most likely to vocalise after receiving rice (Positive + Low Arousal) and rice coated with 4% quinine (Negative + High Arousal).

To explore whether the probability of vocalising could indicate valence or arousal, vocal activity was pooled from the negative stimulus types quinine and air puff, and we ran a generalised linear model with a binomial distribution. The stimulus’ arousal and valence values (dummy coded) and the interaction of arousal and valence were used as predictors. It was found that the valence of stimuli and the interaction between arousal and valence of stimuli influenced the probability of vocalising (Figure 11); however, the model’s overall explanatory power with only valence, arousal, and valence ×x arousal as predictors was very weak (Tjur’s R^2^ = 0.02). The model’s intercept, corresponding to negative valence and low arousal, was at 0.12 (95% CI [−0.03, 0.27], *p* = 0.130). Within this model, the effect of valence was statistically significant and positive (beta = 0.18, 95% CI [0.03, 0.33], *p* = 0.021). The effect of arousal was statistically non-significant and negative (beta = −0.07, 95% CI [−0.22, 0.08], *p* = 0.361). The valence × arousal interaction was statistically significant and negative (beta = −0.23, 95% CI [−0.39, −0.08], *p* = 0.003). Post hoc comparisons of the interaction showed that hens were 0.56 times less likely to vocalise in low-arousal negative compared to low-arousal positive conditions (*p* = 0.0021, Table 3). Hens were also 1.83 times more likely to vocalise in high-arousal positive than low-arousal positive conditions (*p* = 0.015, Table 3).

The probability of vocalisations exhibiting variation across the time of day and over experimental days was also evaluated. A generalised linear model with a binomial distribution was fitted to predict the probability of vocalising with experimental day, time (divided into 2 h bins from 08:00–22:00), and trial number. The model’s explanatory power was weak (Tjur’s R^2^ = 0.08) but showed that the probability of vocalising varied depending on the experimental day, time of day, and trial. The model’s intercept, corresponding to the first day, timebin 8 AM, and trial 1, was at 0.36 (95% CI [0.17, 0.55], *p* < 0.001). Within this model, the effect of day [first degree] was statistically significant and positive (beta = 10.66, 95% CI [6.14, 15.25], *p* < 0.001). The effect of day [second degree] was statistically significant and positive (beta = 6.69, 95% CI [2.50, 10.97], *p* = 0.002). The effect of timebin [first degree] was statistically significant and negative (beta = −10.38, 95% CI [−14.58, −6.23], *p* < 0.001). The effect of timebin [second degree] was statistically significant and negative (beta = −8.90, 95% CI [−13.05, −4.78], *p* < 0.001). The effect of trial was statistically significant and negative (beta = −0.71, 95% CI [−1.01, −0.41], *p* < 0.001). Thus, the probability of vocalising increased over experimental days and was higher in the morning and early afternoon than in the evening (Figure 12). This indicates a possible inverse relationship between feeder engagement and vocal activity. During the day, the hens used the feeder less but had a greater probability of vocalising. Similarly, over experimental days, the hens used the feeder less often but were more likely to vocalise when they did interact with it. The probability of vocalising also varied by trial, with the hens in trial 1 having 2.13 greater odds of vocalising than the hens in trial 2.

#### 3.2.2. Amount of Vocalisations

This study further investigated whether the amount of vocalisations differed depending on the stimulus that the hens received. Based on visual inspection, hens vocalised more in response to positively valenced stimuli; however, this may have been caused by the larger number of files in the positive conditions compared to the negative conditions (Figure 13A). To account for the unbalanced sample sizes, a random selection of 50 times the feeder delivered each valence and arousal combination (Positive + Low Arousal, Positive + High Arousal, Negative + Low Arousal, Negative + High Arousal, total 200 files) was made. The number of vocalisations made in each 1 min recording was then counted. Consecutive vocalisations were counted as a single vocalisation if they were separated by less than 2 s of silence. Plotting the number of vocalisations for these 50 selected files shows that hens seem to vocalise more to negative than positive stimuli (Figure 13B).

To assess this difference statistically, the glmmTMB package was used to run a generalised linear mixed model with a Generalised Poisson distribution on the number of vocalisations. Stimulus valence, arousal, and valence × arousal interaction were included as fixed predictors in the model. The model’s explanatory power was weak (pseudo-R-squared calculated with MuMIn package = 0.042). The model’s intercept, corresponding to valence = negative and arousal = low, was at 1.37 (95%CI [1.33, 1.42, *p* < 0.001). Within this model, arousal was the only significant predictor, with hens vocalising significantly more to high- than low-arousal stimuli (beta = 0.06, 95% CI [0.02, 0.11], *p* = 0.001, Figure 14). The effect of valence was statistically non-significant and negative (beta = −0.04, 95% CI [−0.08, 0.002], *p* = 0.065). The effect of valence x arousal was also statistically non-significant and positive (beta = −0.02, 95% CI [−0.06, 0.02], *p* = 0.271).

## 4. Discussion

To the best of the authors’ knowledge, the present study is the first to explore hens’ voluntary interaction with an ACI device and their vocal activity under evoked emotional arousal and valence states. The results show that the proposed design motivated hens to engage with the feeder frequently despite receiving both positive and negative stimuli, and they only showed a gradual decline in interest after more than a week in one of the trials. In trial 1, regular feed was provided upon the vast majority of activations, with the birds engaging with the device even in those cases, suggesting that the hens’ interest was not entirely due to the presence of novel food rewards (mealworms and rice). This finding is aligned with previous suggestions that animals have an intrinsic drive to explore that goes beyond foraging for resources [38,39]. The engagement levels observed in both trials highlight the ACI device’s potential as an effective environmental enrichment tool for hens. Enrichment strategies that enhance environmental complexity and provide opportunities for voluntary exploration have been shown to stimulate investigation and reduce neophobia [40]. The nature of the device, which encourages hens to approach and interact with it, is aligned with such strategies and might have beneficial effects on hens’ welfare by reducing excessive fear, enhancing cognitive capacity, and improving adaptation skills [41].

The hens’ feathers absorbed the ultrasonic waves, leading to a situation in which a bird could be less than 10 cm from the sensor and not trigger it. The system only triggered when the ultrasound hit featherless areas like the hens’ legs or head. Nonetheless, the hens quickly learned to activate the system with their head in just a day (multiple activations were detected; Appendix A). One possible explanation for this behaviour was that the sensor itself produces sound within the hens’ audible range. Some hens even started pecking on the sensor. However, this meant that the hens were not positioning their head directly above the feeder’s baseline container upon feeder activation. Thus, hens did not receive an air puff to the face, but were startled by the loud and sudden noise of the puff.

The movement or sound of the disc’s rotation seemed to attract the hens’ interest. Thus, the speaker was mostly useful for defining the onset and offset of feeder activations in sound analysis and labelling (see Section 2.3.2), rather than attracting hens to the feeder. Disc movement also posed a problem, as hens started stepping on the disc, which disrupted the rotating process. Remarkably, the hens even managed to intentionally turn the disc to reach containers with mealworms (Appendix A). Pigeon sticks taped on the surface of the disc successfully prevented such behaviours.

The hens quickly learned how to activate the sensor to obtain preferred treats, although there was a noticeable difference between the two trials (averaging 244 activations daily in trial 1 and 72 in trial 2). In both trials, their usage persisted over a week, demonstrating peaks and troughs despite the introduction of mild punishments such as air puffs and distasteful food. The following discussion will explore how variations between the two trials could be due to differences in the probability of reward, the inclusion of mild negative stressors, and individual differences.

Stimuli were dispensed randomly, but with varying probabilities across trials. Trial 2, in which mealworms were provided 85% of the time, had 33% less activations than trial 1, in which mealworms were presented 5% of the time, and the hens’ interest seemed to be shorter. Such results suggest faster habituation when the reward probability is high. This aligns with the observations in [42], where it was noted that because animals have adapted to the unpredictability inherent in their natural environments, the most effective enrichment strategies are likely to involve rewards that are varied and unexpected. The lower reward probability of mealworms and rice in trial 1 could have increased the hens’ effort to obtain them, thus increasing the time it took for hens to habituate to the feeder. When that effort was minimal and mealworms were often provided in trial 2, the hens showed decreased engagement after a week. Longer trial durations would be needed, with higher stimulus uncertainty, to fully understand the scope of this phenomenon.

While the negative effects of distress have been widely recognised, the authors of [43,44] introduced the concept of ‘eustress’ to describe the enhancement of the body’s natural responses to stressors that are not disruptive to an animal’s homeostasis. Reference [45] also suggests that introducing positive stressors, which also alter the arousal of animals but in a beneficial manner, could enhance animals’ cognitive functioning. This is reflected in our findings, where the integration of mild, non-harmful punishments in trial 1 led to increased feeder engagement. In contrast, trial 2, without negative food stimuli, had fewer activations. The probability of encountering negative stimuli was 10% in the first trial and 5% in the second. Despite the startling effect of the air puffs, the hens continued to actively use the feeders while showing high interest in consuming the mealworms, suggesting that they were willing to risk being startled to acquire preferred foods. Future trials could eliminate negative stimuli to investigate how these mild punishments contribute to the overall interest in the device.

Distinct differences in behaviour among individual birds were observed, particularly in terms of curiosity, inventiveness, and food preferences, which likely contributed to the varying levels of engagement. All hens were highly motivated to consume mealworms, but those in the first trial demonstrated a remarkable determination to access them. For example, a hen manipulated the feeder by manually rotating the disc with her feet to access the mealworms. This behaviour points to varying degrees of exploration and curiosity among the birds, with some showing a heightened eagerness to interact with the feeder. Such differences can be explained with respect to the proactive–reactive axis [46], with proactive or bold birds showing a higher tendency to explore their environment, while reactive or shy hens are more cautious with external objects [47].

Regarding this study’s long-term objective, data from the first two trials indicate the device’s efficacy in eliciting vocalisations, although the probability of vocalising did not reliably distinguish between emotional valence and arousal. Birds vocalised slightly more than half of the time (51%) during feeder activation. Interestingly, the daily and diurnal patterns in feeder usage did not match the probability of vocalisations or the number of vocalisations. The amount of vocalisations was positively related to emotional arousal, aligning with expectations, and suggests that stimuli were effective at inducing the intended level of arousal. Nonetheless, the measured indices, i.e., the probability of vocalising and the number of vocalisations, were not reliable indicators of emotional valence. Further analysis of the vocalisations’ properties would be beneficial to investigate differences across valence states. It has been proposed [7] that the duration and the frequency distribution of the spectrum of the calls, as well the type of the call, can help in distinguishing the valence of animals. Other behavioural and physiological data would help to further validate that hens’ arousal levels were indeed higher or lower as anticipated by the stimuli. Moreover, expanding this study to include a larger sample of birds would be essential for making more definitive statements about emotional valence and arousal in chicken vocalisations.

Another unexpected finding was the inverse correlation between feeder activations and the likelihood of vocalisations across experimental days. Staring with a relatively high probability, perhaps due to the novelty of the environment, the hens‘ engagement increased while the probability decreased. Over the last experimental days of each trial, the probability reached the maximum levels, while engagement was either constant or slightly decreased. Our speculation on this interesting result is that as the hens became familiar with the provided stimuli, they increasingly produced anticipatory calls when facing a reward [48] and frustration-related calls when negative stimuli were provided or when not receiving rewards [49]. Future studies should identify the call types made in various reward and non-reward conditions and examine whether they increase over time. The same inverse relationship was observed within experimental days, with birds being slightly more likely to activate the feeder towards the evening, while the probability of vocalising was at the lowest level during the same time. Natural behaviours could explain this phenomenon, with birds being less vocally active during the evening and more active during the day. Morning hours often include egg-laying and feeding activities [50,51], during which hens are more likely to produce vocalisations. Analysing the spontaneous vocal activity of hens could support this interpretation.

One potential issue with the existing setup involves the competition of birds over access to the provided stimulus, mainly the mealworms and rice. As observed in [40], hens can be highly competitive even when their nutritional needs are fulfilled. This was the case mainly in trial 1 and to some extent for trial 2, where dominant hens often monopolised access to both feeders, using them even when the subordinate hen was the one that activated the apparatus (Appendix A). Moreover, in the absence of direct competitive behaviours, it was observed that birds often used the feeder simultaneously. Such events can lead to ambiguous emotional states and, consequently, vocalisations that cannot be assigned to a single context, since one bird might enjoy the provided mealworms while the other bird produces sounds related to frustration and negative valence. This study did not specifically focus on identifying which bird activated the feeder or produced a vocalisation; however, this could further enhance our understanding of individuals’ emotional states. The integration of RFID sensors, coupled with a proximity sensor to allow only one individual to activate a feeder, might be one way to overcome this problem. This could reduce the undesired effect of simultaneous usage but not entirely eliminate it, since the other hen could easily try to reach the feeder’s content after the activation. Another possible way to bypass this could be the assignment of activations and vocalisations to individual birds, either algorithmically or by inspecting the video data.

## 5. Conclusions

This study introduced an interactive feeder designed to elicit vocalisations from laying hens in different emotional valence and arousal states. The active engagement of the hens with the feeder over several days highlights its potential as an environmental enrichment tool. The birds showed motivation to interact with the device, which dispensed both positive and negative stimuli, leading to a substantial collection of vocalisations. A preliminary vocalisation analysis indicated that birds vocalised more in states of high arousal compared to low arousal. However, the number of vocalisations did not significantly differentiate between positive and negative valence states. This study lays the groundwork for a deeper exploration into the emotional states of poultry using vocal expression and underscores the potential of the presented ACI device in animal welfare research.

## Figures and Tables

**Figure 1 animals-14-01386-f001:**
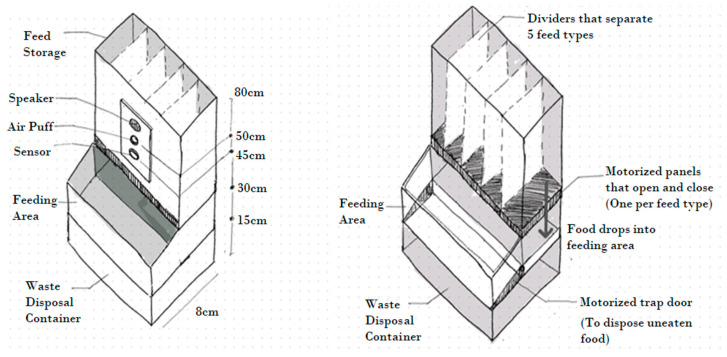
The tower design.

**Figure 2 animals-14-01386-f002:**
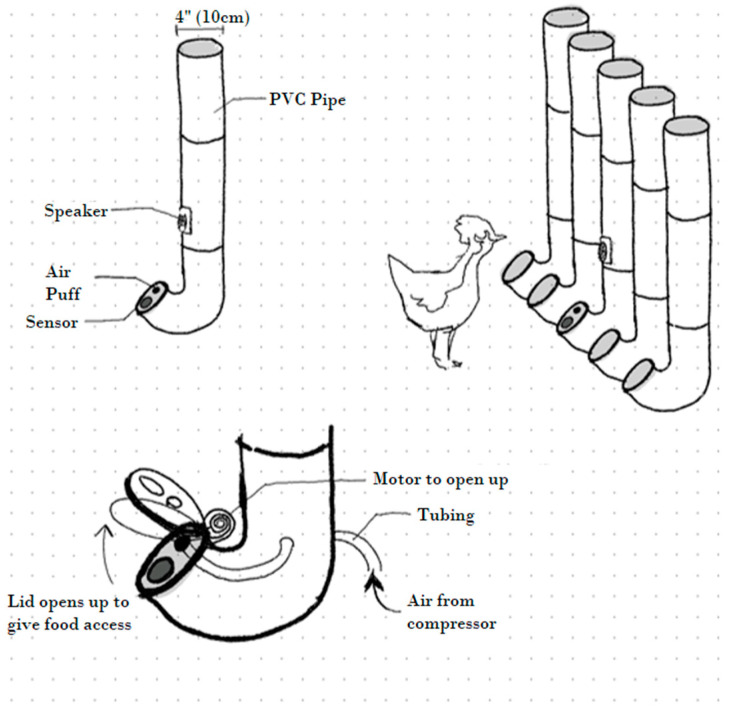
The tube design.

**Figure 3 animals-14-01386-f003:**
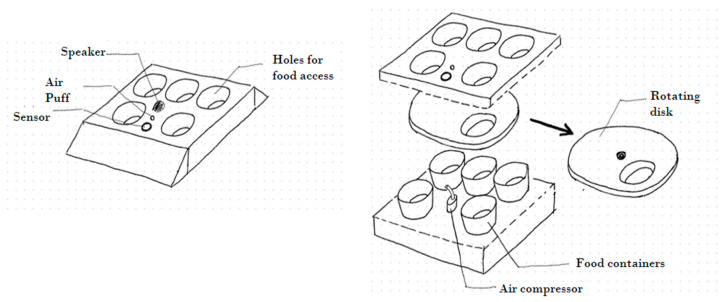
The floor rotator design.

**Figure 4 animals-14-01386-f004:**
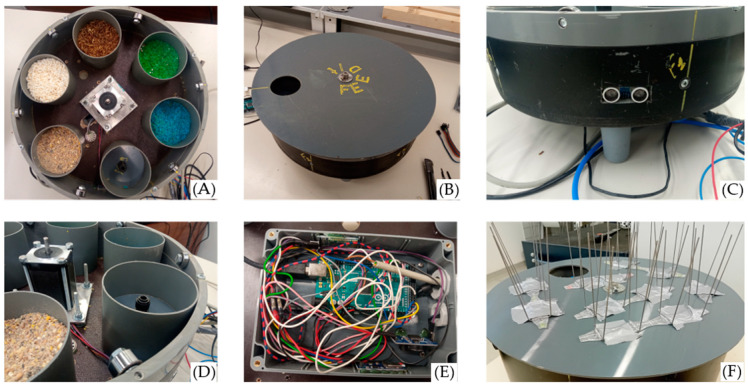
The proposed interactive feeder. (**A**) The food containers, (**B**) the rotating disc, (**C**) the ultrasonic sensor on the side of the device, (**D**) metallic wheels aiding the rotation, (**E**) the box containing the electronics, positioned on the bottom, and (**F**) the final form of the feeder with pigeon spikes taped on the disc.

**Figure 5 animals-14-01386-f005:**
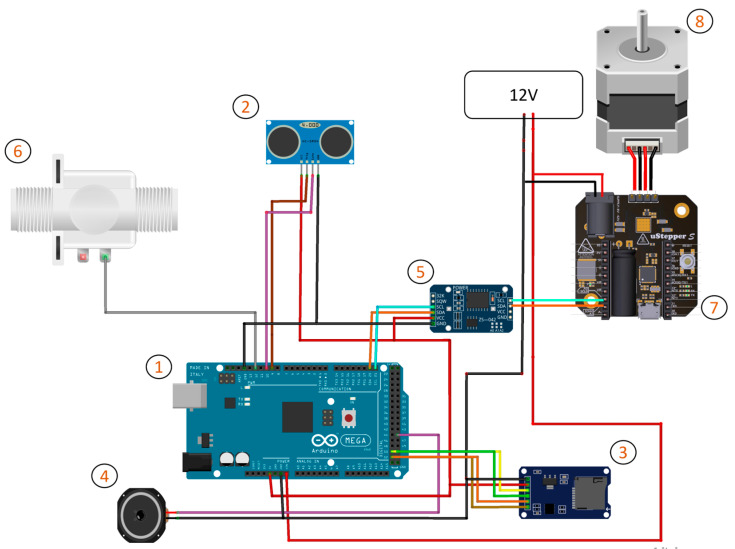
System’s block diagram. Hardware components: (1) Arduino Mega, (2) Ultrasonic Sensor, (3) Real-Time Clock, (4) miniature speaker, (5) Micro-SD card reader, (6) pneumatic solenoid valve, (7) uStepper S stepper driver, and (8) Nema 23 stepper motor.

**Figure 6 animals-14-01386-f006:**
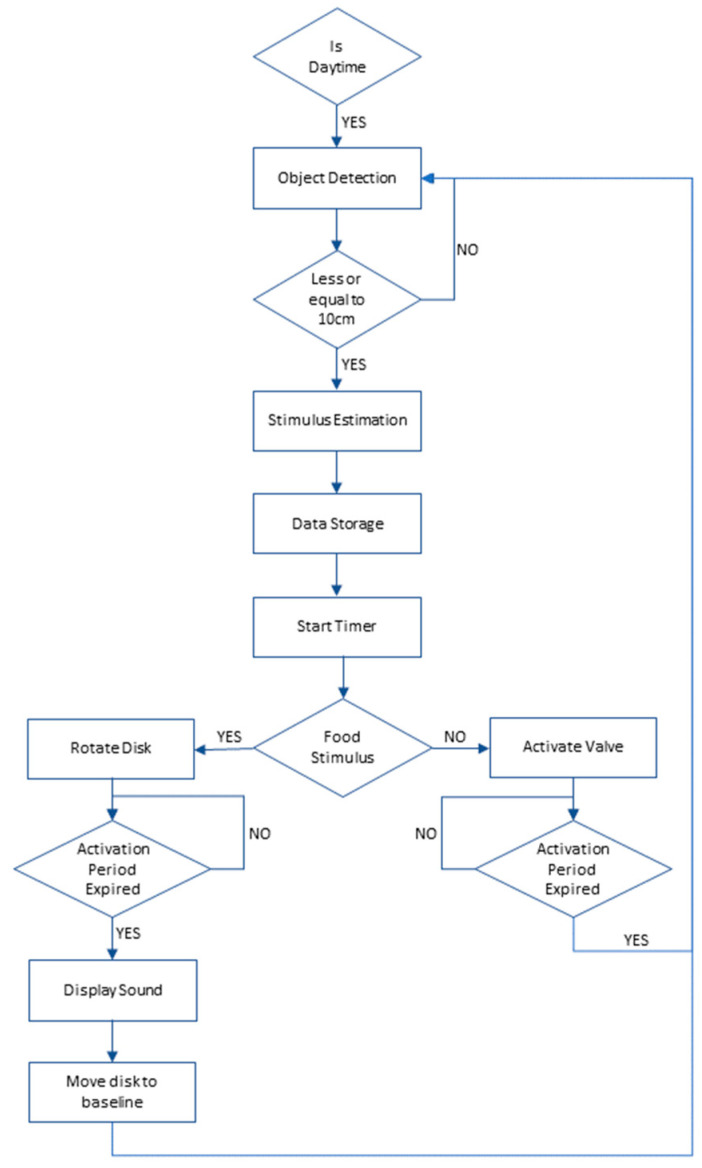
The flowchart of the proposed algorithm.

**Figure 7 animals-14-01386-f007:**
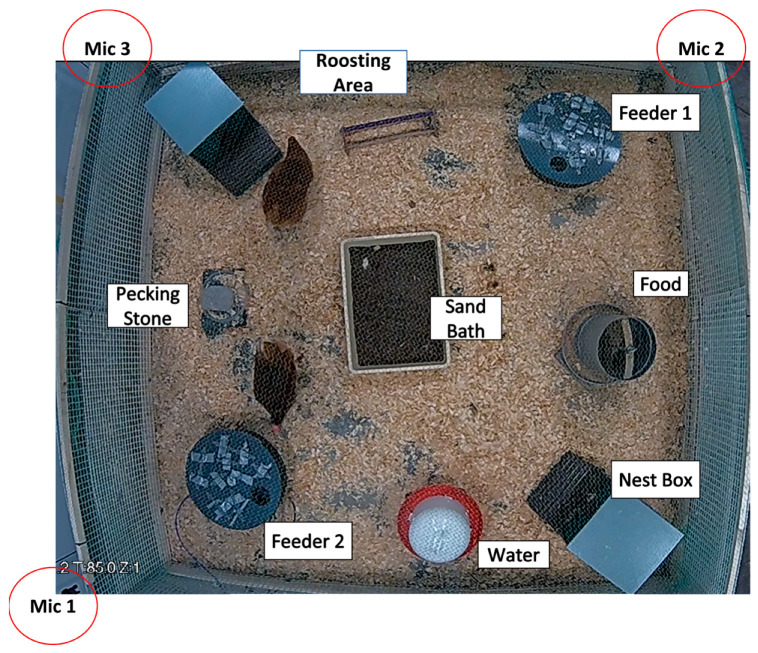
The pen and placement of recording devices.

**Figure 8 animals-14-01386-f008:**
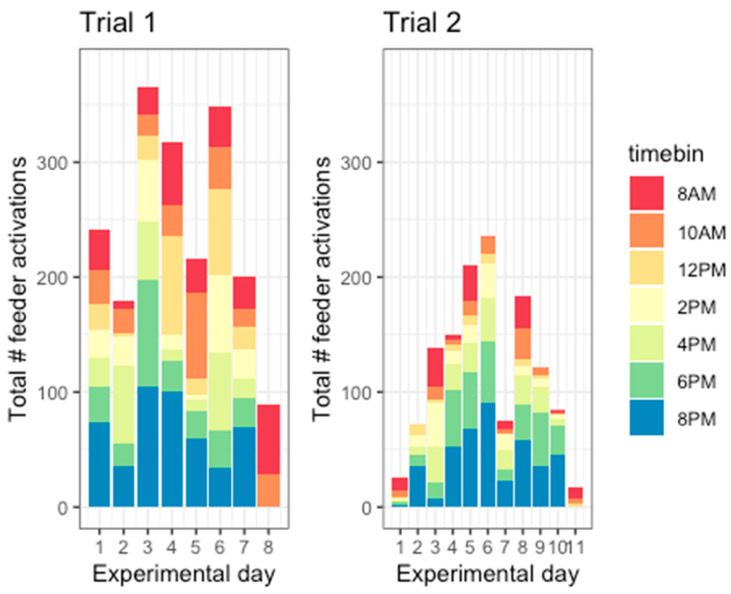
Number of times feeders were activated in trial 1 and trial 2 across all experimental days. Colours indicate time of day, divided into 2 h bins.

**Figure 9 animals-14-01386-f009:**
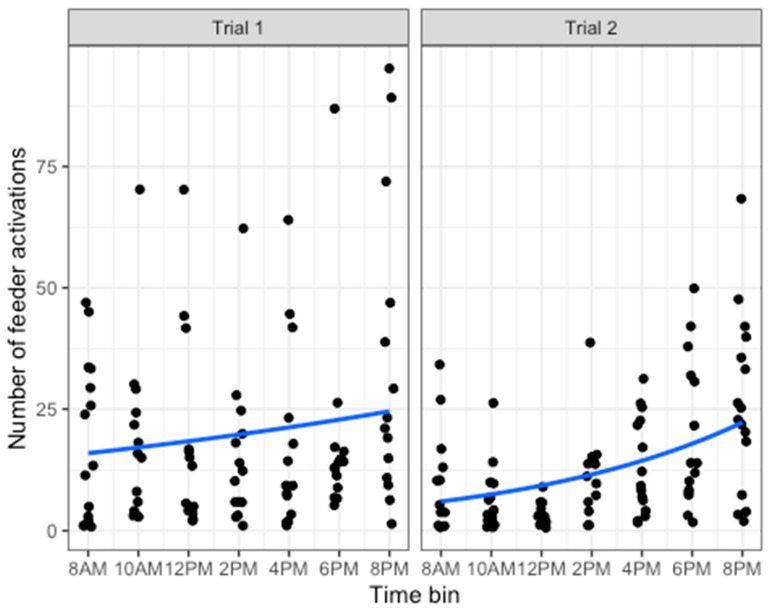
Number of times feeder activated at different times of the day. Time of day was divided into 2 h bins. Dots represent an activation. Blue lines show the trend across time of day.

**Figure 10 animals-14-01386-f010:**
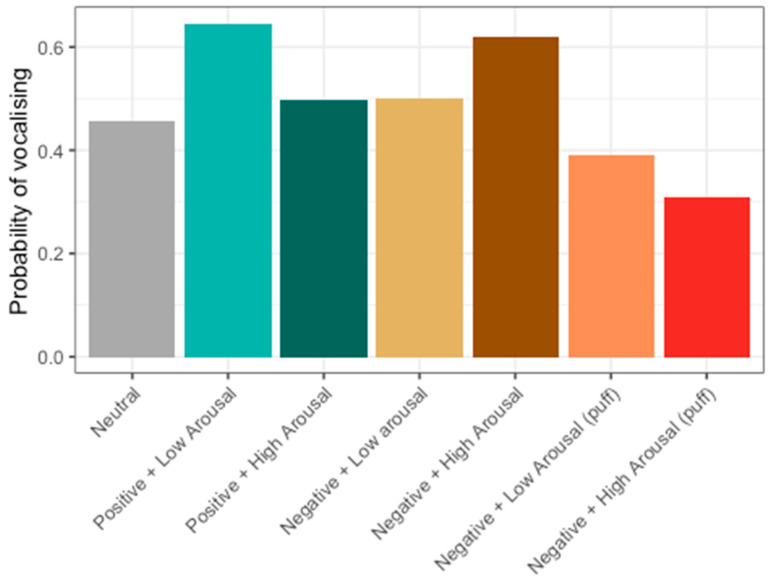
Probability of hens vocalising after receiving each stimulus type.

**Figure 11 animals-14-01386-f011:**
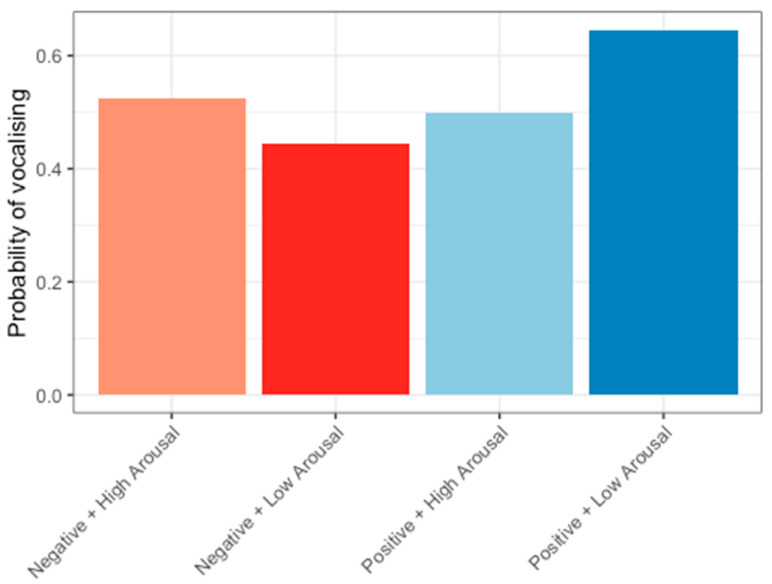
Probability of vocalising in response to stimuli varying in valence and arousal. The odds of hens vocalising was greater in positive low-arousal conditions than negative low-arousal and positive high-arousal conditions.

**Figure 12 animals-14-01386-f012:**
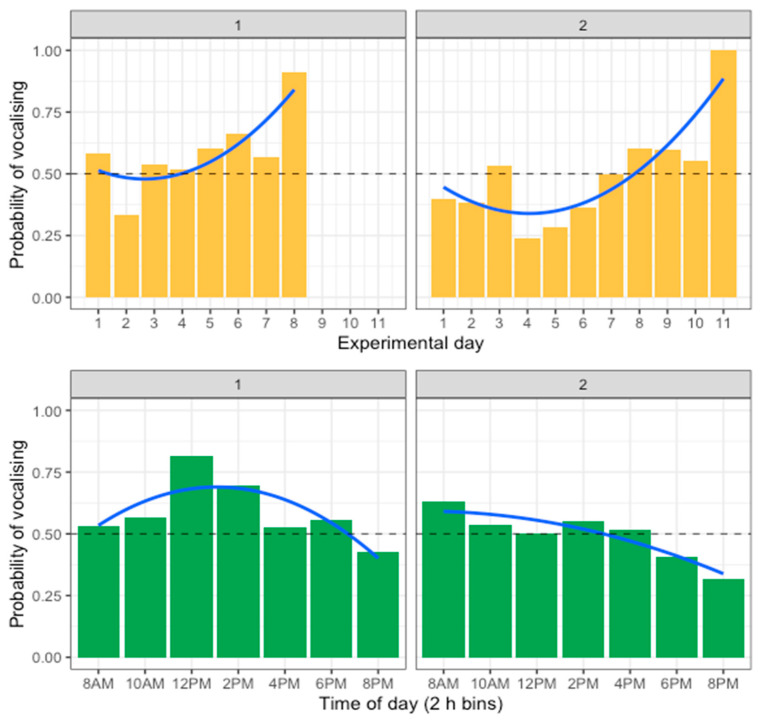
Change in probability of vocalising over experimental days and time of day, separated for trial 1 and trial 2. Lines show the trend across days and time of day.

**Figure 13 animals-14-01386-f013:**
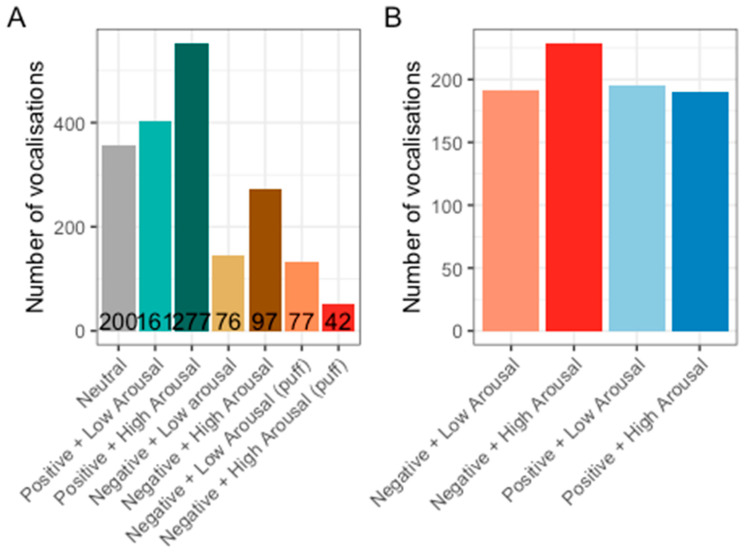
Number of vocalisations in response to the different stimuli types (**A**) and in response to 50 deliveries of the four different valence and arousal combinations (**B**). Numeric values in (**A**) indicate the number of files analysed per stimulus type.

**Figure 14 animals-14-01386-f014:**
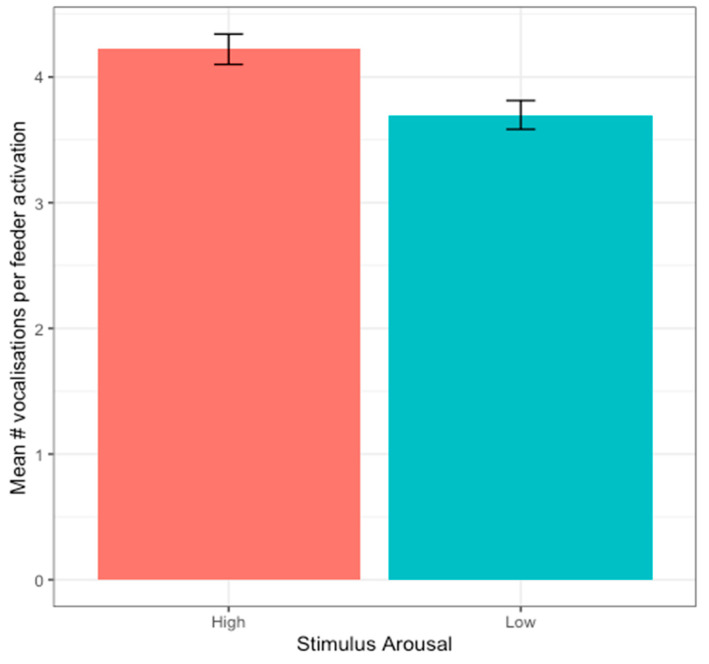
Number of vocalisations in response to stimuli with positive or negative arousal.

**Table 1 animals-14-01386-t001:** List of experimental stimuli and their assigned valence and arousal conditions.

Stimulus	Valence	Arousal
Regular Feed	-	-
Mealworms	Positive	High
Rice	Positive	Low
1% Quinine Suspension	Negative	Low
4% Quinine Suspension	Negative	High
Two air puffs	Negative	Low
Four air puffs	Negative	High

**Table 2 animals-14-01386-t002:** Advantages and disadvantages of the proposed design ideas.

Design	Advantages	Disadvantages
Tower	− Good thermal camera view − Low risk of food spilling	− Complex mechanism for food weight − Requires six motors − Mixed uneaten food leading to waste − Questionable accessibility by hens
Tube	− No need to remove uneaten food − Partially tested by farmers − Good thermal camera view	− Challenging motorised lid construction − Risk of disruption − Potential blockage of air puff
Floor Rotator	− No need to remove uneaten food − Mimics natural foraging − Single motor required − Hard to disrupt	− Potentially poor thermal camera view − Complex elevated platform construction

**Table 3 animals-14-01386-t003:** Contingency table showing the number of times hens vocalised or did not vocalise after activating the feeder.

Condition	Negative + Low Arousal	Positive + Low Arousal	Negative + High Arousal	Positive + High Arousal
Vocalise_Yes	85	57	66	139
Vocalise_No	68	104	73	138

## Data Availability

The original contributions presented in the study are included in the article and Appendix A, further inquiries can be directed to the corresponding authors.

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
