# Peer review of "An Interactive Feeder to Induce and Assess Emotions from Vocalisations of Chickens"

_animals, 2024, doi:10.3390/ani14091386_

Round 1

Reviewer 1 Report

Comments and Suggestions for Authors

Line 187, 215, 244: I suggest the consideration of creating digital schematics for the figures currently presented as scanned hand-drawn images. Utilizing digital renderings can enhance the visual clarity and professional appearance of these figures.

Introduction

Clearly outlines the purpose of the study and its relevance in the field of animal welfare, specifically focusing on hens.

Materials and methods

This section provides a thorough description of the materials and methods used, which is crucial for replicability in scientific research.

Results

Clear and structured. The inclusion of charts, graphs and tables, enhances the presentation of the data.

Discussion

Ensure that the discussion maintains a balanced perspective, acknowledging any limitations of the study and how they might impact the results.

Conclusion

I have a query regarding the conclusion of the paper, which states that “vocalization analysis indicated that birds vocalized more in states of high arousal…the number of vocalizations did not significantly differentiate between positive and negative valence states”. This statement appears to be somewhat at odds with the title of the paper “Assessing positive welfare”. Given this discrepancy, it might be worth reconsidering the title to reflect the findings of the study more accurately.

Reviewer 2 Report

Comments and Suggestions for Authors

The paper describes an attempt to develop a methodology to remotely assess positive and aversive experiences in hens. It is innovative and laudable, but unfortunately does not meet appropriate scientific standards for publication in a peer-reviewed journal. The paper presents the results of a study that could be best described as an un-replicated pilot study.

The following points describe the most serious flaws of the study:

1.            Careless use of the scientific literature. For example, in the Introduction the authors reported that their study was inspired by two papers (Refs 16 and 17) that analysed hen vocalisations during anticipation of events varying in valence. However, Goldfidis et al. measured vocalisations during the presentation of different events and there was no mention of how variations in the temporal relationship between stimulus presentation and behaviour might influence the hen’s experiences.

2.            Conceptual/design deficiencies. The study assumed that stimuli that are voluntarily approached (positive stimuli) would elicit positive emotional responses and those that are avoided (aversive stimuli) would elicit negative emotional responses (Lines 118-122). However, the methodology used to deliver both positive and negative stimuli and measure the responses required the hens to approach all types of stimuli. Further, the probability of encountering a positive event was many times higher than that for aversive events. This means that approaches to the negative events were likely due to the high probability of encountering a positive event. Thus, from the hen’s point of view, the values of the positive and aversive stimuli are likely to have been conflated. There is ample evidence in the behaviour analytic literature for this type of stimulus conflation by animals (e.g.Weiss, S. J. (1996). Combining operant baseline derived conditioned excitors and inhibitors from the same and different incentive classes: an investigation of appetitive-aversive interactions. The Quarterly Journal of Experimental Psychology: Section B49(4), 357-381; Cohn, S. I. (2002). Compounding olfactory and auditory discriminative stimuli on a free-operant baseline. American University).

3.      Additional methodological flaws relate to the too few numbers of experimental animals used (four in total), lack of visual and auditory isolation of the birds within and between pens (which means the birds were not independent experimental units) and pooling of data across both Trials even though the stimulus presentation regimes differed significantly between them.

 4.      All sections of the paper are overly long, and (especially) the Introduction section lacks cohesion.

Comments on the Quality of English Language

All sections of the paper are overly long, and the Introduction section, in particular, lacks cohesion and logical flow.

Round 2

Reviewer 1 Report

Comments and Suggestions for Authors

If it is not feasible to improve the diagrams with graphic design software, it is suggested to improve the legends of each illustration computationally. it is important to search for basic and easy to use design program as a tool for future studies.
